# The Influence of Pool-Riffle Morphological Features on River Mixing

**Patricio Fuentes-Aguilera** [1,*] , **Diego Caamaño** [1] , **Hernán Alcayaga** [2] **and Andrew Tranmer** [3]

1   Department of Civil Engineering, Universidad Católica de la Santísima Concepción, Concepción 409541, Chile; dcaamano@ucsc.cl
2   Escuela de Ingeniería en Obras Civiles, Universidad Diego Portales, Santiago 8370109, Chile; hernan.alcayaga@udp.cl
3   Center of Ecohydraulics Research, University of Idaho, Moscow, ID 83844, USA; andyt@uidaho.edu
*   Correspondence: pfuentes@magister.ucsc.cl

**Abstract:** Accurate prediction of pollutant concentrations in a river course is of great importance in environmental management. Mathematical dispersion models are often used to predict the spatial distribution of substances to help achieve these objectives. In practice, these models use a dispersion coefficient as a calibration parameter that is calculated through either expensive field tracer experiments or through empirical equations available in the scientific literature. The latter are based on reach-averaged values obtained from laboratory flumes or simple river reaches, which often show great variability when applied to natural streams. These equations cannot directly account for mixing that relates specifically to spatial fluctuations of channel geometry and complex bed morphology. This study isolated the influence of mixing related to bed morphology and presented a means of calculating a predictive longitudinal mixing equation that directly accounted for pool-riffle sequences. As an example, a predictive equation was developed by means of a three-dimensional numerical model based on synthetically generated pool-riffle bathymetries. The predictive equation was validated with numerical experiments and field tracer studies. The resulting equation was shown to more accurately represent mixing across complex morphology than those relations selected from the literature.

**Keywords:** dispersion coefficient; fluid mixing; dimensional analysis; pool-riffle; 3D modeling

## 1. Introduction

Understanding the fate and transport of introduced pollutants and substances within river courses is relevant for public health, ecological diversity, and the administration of water resources [1–4]. Fundamental to this understanding is the accurate prediction of substance concentrations and their distribution, owing to mechanisms such as advection, molecular diffusion, and dispersion, where the former is the most dominant process in natural rivers [5]. Dispersion in the longitudinal, lateral, and vertical directions accounts for the effects of spatial differences in velocities over the channel cross-section, and consequently its magnitude depends upon the scales of turbulent diffusion and mixing, owing to channel irregularities [6–10]. Prediction of longitudinal mixing is complicated in natural rivers as the channel morphology increases in complexity (e.g., planform curvature, bed irregularity, variable roughness provided by macroforms, substrate, and vegetation) [11–16]. Under such circumstances the inertial terms in the hydrodynamic equation become increasingly important for mixing and pollutant transport. Therefore, the morphological features within a natural

stream reach, such as contractions, expansions, and bed macroforms, must be accounted for in the fate and transport of dissolved and suspended constituents.

Different mathematical tools are currently used to characterize these highly complex three-dimensional (3D) mixing processes, all of which are based on the three-dimensional advection–dispersion–reaction equation (Equation (1)).

$$\frac{\partial C}{\partial t} = u\frac{\partial C}{\partial x} + v\frac{\partial C}{\partial y} + w\frac{\partial C}{\partial z} + \frac{\partial}{\partial x}D_x\frac{\partial C}{\partial x} + \frac{\partial}{\partial y}D_y\frac{\partial C}{\partial y} + \frac{\partial}{\partial z}D_z\frac{\partial C}{\partial z} - R \tag{1}$$

where $t$ represents time; $C$ is the concentration of the substance; and $u$, $v$, and $w$ correspond to the velocities in the directions $x$, $y$, and $z$. $D_x$, $D_y$, and $D_z$ are the dispersion coefficients in each associated direction. The term $R$ represents the sources and sinks that consume or contribute mass to the system (i.e., null value in the case of a conservative substance). The velocity field values ($u$, $v$, $w$, $t$) are obtained by solving hydrodynamic equations (i.e., Saint-Venant equations, or variant equations) and the dispersion coefficients ($D_x$, $D_y$, $D_z$, $t$) characterize the directional mixing that are necessary to complete the mathematical scheme [11,17–19].

Therefore, to solve this system of equations, the necessary data requirements are high, given the complexity of the numerical solution when coupling both the hydrodynamic and transport processes. In practice, several simplifications can be made to reduce the complexity of Equation (1) [6,20–25]. For example, it is frequently assumed that mixing in the vertical and transverse directions occurs instantaneously, which allows the estimation of the time–space variation of the concentration in the longitudinal direction, via a one-dimensional (1D) model [26,27]. Several empirical equations are available in the literature following this one-dimensional simplification and a sample of them is listed in Table 1. None of the equations in Table 1 explicitly account for channel irregularities or morphological complexity outside of general channel dimensions.

**Table 1.** One-dimensional longitudinal dispersion formula-associated simplifications.

| Reference | Formula | Simplifications |
|---|---|---|
| Elder [28] | $D_x = 5.93Hu_*$ | Uniform flow in an infinitely wide channel. |
| Fischer [29] | $D_x = 0.011\left(\frac{B^2}{H}\right)\left(\frac{U^2}{u_*}\right)$ | Validated using measurements in straight prismatic channels of various regular cross-sectional shapes. |
| Seo and Cheong [26] | $D_x = 5.915\left(\frac{B}{H}\right)^{0.620}\left(\frac{U}{u_*}\right)^{1.428}Hu_*$ | Developed using dimensional analysis and the one-step Huber method [30]. |
| Kashefipour and Falconer [31] | $D_x = 10.612\left(\frac{U}{u_*}\right); \ for \ B/H > 50$  $D_x = \left[7.428 + 1.775\left(\frac{B}{H}\right)^{0.620}\left(\frac{u_*}{U}\right)^{0.572}\right]HU\left(\frac{U}{u_*}\right); \ for \ B/H < 50$ | Calibrated and validated using data from 30 streams in USA; previously used by Fischer [32], McQuivey and Keefer [33], and Seo and Cheong [26]. |
| Zeng and Huai [34] | $D_x = 5.4\left(\frac{B}{H}\right)^{0.7}\left(\frac{U}{u_*}\right)^{0.13}HU$ | Calibrated and validated using data from 50 rivers in the USA. |
| Sahin [35] | $D_x = \beta R_h U$ | Developed using dimensional analysis. This equation includes the hydraulic radius and the shape of the cross-section |

To obtain these estimates of reach-averaged dispersion coefficients, many authors have carried out tracer studies [26,33,36–41]. Many of these empirical models have been validated in the laboratory or simple stream reaches, whereas natural alluvial rivers present variations in planform and bed topography, which deviate from a plane configuration [4] and rarely satisfy the implicit assumptions (Table 1). Therefore, these empirical equations offer limited use because their application is only valid in scenarios that are similar to those used in their derivation. Predicted longitudinal dispersion for the same river reach might vary widely depending on the equation used [26,35,42,43]. This stems from

various local morphological complexities, such as step-pools, sinuosity, and pool-riffle macrostructures being lumped into one value of reach-averaged longitudinal dispersion. When the distribution of specific morphological features like pool-riffle macrostructures (i.e., residual pool depth, riffle height, length of bars, widths, etc.) differ from one reach to another, the resultant dispersion coefficient might be less appropriate. Hence, there is still uncertainty regarding the best method for representing the complex three-dimensional mixing processes in a single, one-dimensional parameter, especially in non-uniform channels [43].

This study accounts for morphological complexities arising directly from pool-riffle macrostructures within a river channel, which results in improved predictions of the longitudinal dispersion coefficient. A three-dimensional coupled hydrodynamic and transport model that includes turbulence fluctuations and variations in bathymetry was developed to provide output data necessary to characterize the complex velocity field through pool-riffle macrostructures. Ten different synthetically generated pool-riffle bathymetries provide a physical basis for the model runs. The numerical results were subject to a dimensional analysis and an empirical equation capable of predicting the longitudinal dispersion coefficient through pool-riffle structures was proposed for an improved one-dimensional simplification. Five field tracer studies were additionally used to validate the predictive equation.

## 2. Materials and Methods

Formulation and validation of the proposed longitudinal dispersion equation included (i) synthesizing realistic pool-riffle bathymetries to provide different geometric macrostructures, (ii) three-dimensional numerical hydrodynamic and transport modeling in order to obtain target dispersion coefficient values, (iii) testing a series of dimensionless numbers to characterize the given geometry and flow conditions in pool-riffle macrostructures that accurately predict the target dispersion coefficients, and (iv) applying numerical results and field tracer experiments across a range of values to validate the proposed equation, in order to better represent mixing through complex bed macrostructures.

### 2.1. Syntethic Pool-Riffle Generation

A synthetic river channel bathymetry was developed, based on the synthetic river valley (SRV) methodology [44,45] (Figure 1). The method used mathematical functions to generate realistic geometry of longitudinal and transversal synthetic river profiles [45]. Application of this method created a channel with a pool-riffle configuration based on average bankfull width ($\overline{W_{BF}}$), average particle diameter ($\overline{D_{50}}$), average riverbed slope ($\overline{S}$), critical Shields parameter ($\theta_*$), and specific weight of water and solid particles ($\gamma_w$ and $\gamma_s$). SRV bathymetries were defined to represent the average river conditions observed during the field tracer experiments (Section 2.4), with an average bankfull width of 10 m, particle diameter of 0.003 m, and riverbed gradient of 0.002 m/m.

A straight-channel bathymetry was studied in order to observe the effect of the vertical convergence and horizontal divergence of flow characteristics of the pool-riffle macroform on the dispersion coefficient. The elevation of the thalweg and width of the channel were given by separate sinusoidal functions [45] and the transversal and longitudinal shapes were offset (i.e., the pool was located in narrow zones and the riffle in wide zones). Additionally, the cross-sectional shape was based on the parabolic model proposed by Deutsch and Wang [46]. Ten different SRV bathymetry scenarios were defined by incrementally filling in the pool to provide a range of residual pool depths. Residual pool depth was calculated as the difference in channel bed elevation between the thalweg at the deepest section of the pool and the highest point of the riffle [47]. Figure 2 shows the morphology of a channel with a pool-riffle structure, as well as the residual pool depth ($\Delta_z = h_{Pt} - h_{Rt}$), pool width ($B_p$), pool depth ($h_{Pt}$), and riffle depth ($h_{Rt}$). The analyzed SRV residual pool depths ranged from 0.044 m to 0.494 m. Scenario 1 began with $\Delta_z = 0.494$ m and incrementally decreased by 0.05 m (Table 2). The resulting SRV bathymetry scenarios provided the physical boundaries (i.e., closed boundaries) for the numerical modeling.

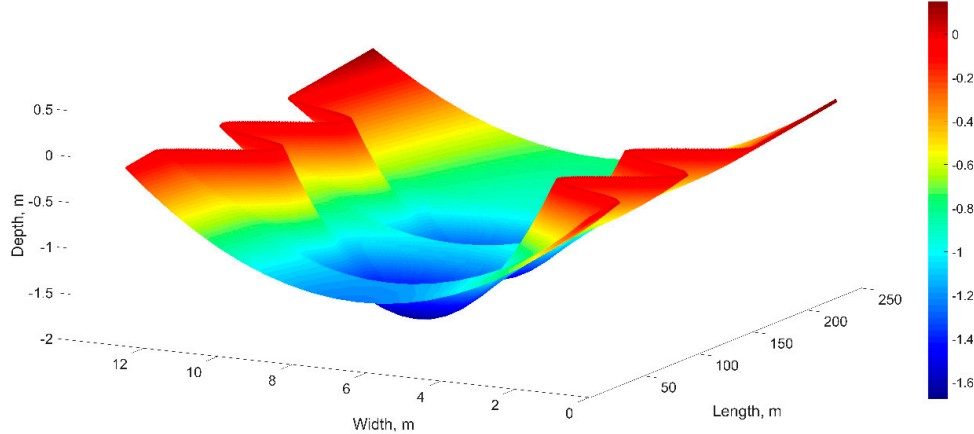

**Figure 1.** Results of the synthetic bathymetry corresponding to scenario 1.

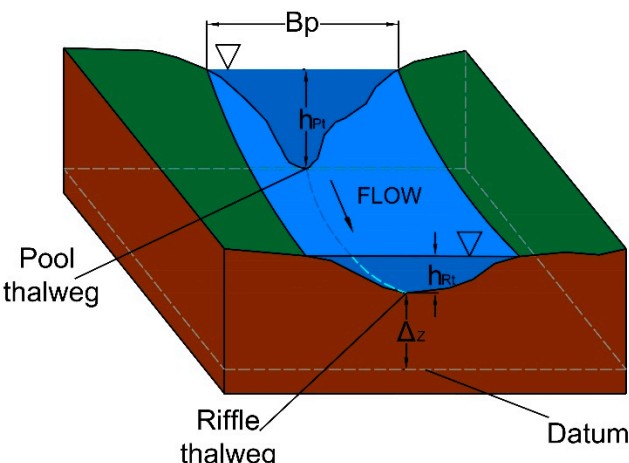

**Figure 2.** Pool-riffle structure and the geometric variables considered in the analysis (modified from Caamaño et al. [47]).

**Table 2.** Considered residual pool depths.

| Bathymetry Scenario | Residual Pool Depth, m |
|:---:|:---:|
| 1 | 0.494 |
| 2 | 0.444 |
| 3 | 0.394 |
| 4 | 0.344 |
| 5 | 0.294 |
| 6 | 0.244 |
| 7 | 0.194 |
| 8 | 0.154 |
| 9 | 0.094 |
| 10 | 0.044 |

### 2.2. Numerical Modelling: Hydrodinamic and Transport Simulations

The hydrodynamic modeling considered in this study solved the three-dimensional Reynolds Average Navier-Stokes equations under the hydrostatic and Boussinesq approximations, using the Delft3D-FLOW model [48]. The numerical solution was obtained by the Alternating Direction Implicit scheme and employing the $k$-$\varepsilon$ turbulence closure method [49]. The solution was calculated on an unstructured grid with rectangular dimensional elements characterized by an average longitudinal value of 0.455 m and a transversal value of 0.140 m. These values of the unstructured grid size were

selected to ensure independence of the model results via a sensitivity analysis. Grid spacing was reduced until no appreciable changes to the flow field were observed. Figure 3 presents the analyses for the cross-sectional average velocity at the head (start), the middle, and the exit (end) of the first pool.

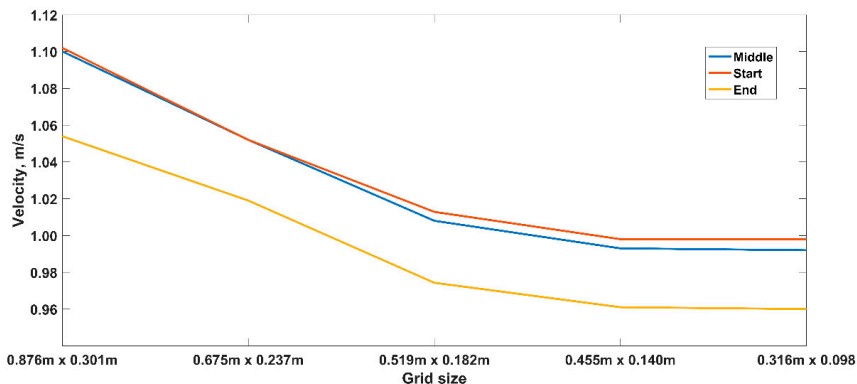

**Figure 3.** Results for the sensitivity analysis for the unstructured mesh size.

The vertical discretization was carried out through the Sigma ($\sigma$) method that uses eight vertical layers according to variations in the flow depth. One numerical simulation was performed for each of the SRV scenarios reported in Table 2. Bankfull discharge was used for the hydrodynamic solution and calculated by the equation proposed by Lee and Choi [50] as 9 m$^3$/s. Channel roughness was modeled using Manning's roughness parameter and was calculated using the Strickler relation as 0.025 (m$^{-1/3}$ s). A singular Manning's roughness parameter was used to isolate the impacts of the pool-riffle bed macro-structure on the velocity field. Discharge increased gradually in the model in order to facilitate stability and convergence toward a stationary hydraulic stage.

The hydrodynamic FLOW model was coupled with the water quality (Delft3D-WQ) model to solve the advection–dispersion–reaction equation [51,52], representing a conservative substance introduced to the flow ($R = 0$ in Equation (1)). The model domain extended 100 m from the upstream riffle where the injection took place, through one pool, to an open boundary condition at the downstream riffle. The conservative constituent was instantaneously injected with its concentration uniformly distributed across the entire riffle cross-section. The uniform distribution at the boundary condition for the water quality model represented an inflow of a constituent introduced sufficiently upstream to allow diffusive mixing through the pool riffle section. Therefore, this provided ideal conditions for evaluating the dominant longitudinal mixing processes. Dispersion processes were assumed to be isotropic ($D_x = D_y = D_z$) and the initial dispersion coefficients were defined as 1 m$^2$/s. Solute transport in the WQ model used the same spatial discretization as the hydrodynamic model. The numerical transport solution was obtained by the minimal residual method [53], which corresponded to an unconditionally stable implicit method.

To characterize the model dispersion coefficient, the water quality numerical results were output at seven observation points with 10 m spacing, located along the centerline of the synthetic pool-riffle sequence to provide concentration-time curves. These concentration-time curves were used to back-calculate the longitudinal dispersion coefficient using Thomann and Mueller's [54] method and the method of moments [55], hereafter termed TM and MM methods. Both methods were widely tested in the literature [6,55–58]. The numerical simulations produced 10 dispersion coefficients that were randomly split into two groups. Seven were used to define the proposed equation and three were used for equation validation. A sensitivity analysis was performed to assess the variability of these results upon the initial dispersion coefficient required to find a solution for the transport model. Thus, sixteen new mixing input sets for scenario 1 were created using isotropic three-dimensional dispersion coefficients with magnitudes ranging from 0.5 to 2 m$^2$/s in increments of 0.1 m$^2$/s. Additionally, a scenario with a null dispersion coefficient was considered, to test for numerical dispersion.

### 2.3. Dimensional Analysis

Dimensional analysis was used to translate the 3D model results into a predictive 1D dispersion coefficient equation for pool-riffle macrostructures. Based on the geometric variables of the synthetic pool-riffle macrostructures and the simulated flow field properties, the longitudinal dispersion coefficient in the study reach was proposed to be a function of the following variables: $D_x = f(\rho,\ \mu, U,\ u^*,\ H,\ S_w,\ S_B)$, where $\rho$ and $\mu$ correspond to the fluid's density and dynamic viscosity; $U, u^*, H$ correspond to average velocity, shear velocity, and depth of the pool-riffle; $S_w$ is the water surface gradient, and $S_B$ is associated with the horizontal expansion between the pool and the downstream riffle. The values of $S_w$ and $S_B$ were calculated according to Equations (2) and (3), respectively.

$$S_w = \frac{h_{Pt} - (h_{Rt} + \Delta_z)}{L_{PR}} \tag{2}$$

$$S_B = \frac{B_r - B_p}{L_{PR}} \tag{3}$$

where $h_{pt}$ represent the depth of the pool, $h_{rt}$ is the depth of the downstream riffle, $\Delta_z$ is the residual pool depth, $B_r$ and $B_p$ are the width of the riffle and pool, respectively, and $L_{PR}$ is the length between the pool and downstream riffle. Dimensional analysis [59] produces four $\pi$-dimensionless numbers represented in Equations (4)–(7). The Kashefipour and Falconer [31] method was used to select the best combination of $\pi$-dimensionless numbers out of 19 possible alternatives (Table 3). The combination with highest correlation value $\left(R^2\right)$ was chosen to define the proposed equation.

$$\pi_1 = \frac{D_x}{u^* \cdot H} \tag{4}$$

$$\pi_2 = \frac{U}{u^*} \tag{5}$$

$$\pi_3 = S_w \tag{6}$$

$$\pi_4 = S_B \tag{7}$$

**Table 3.** Dimensionless combinations in order to improve correlation.

| Combination | Correlation |
|---|---|
| $\pi_1 - \pi_2$ | 0.8383 |
| $\pi_1 - \pi_3$ | 0.8909 |
| $\pi_1 - \pi_4$ | $1 \times 10^{-14}$ |
| $\pi_1 - \pi_2\pi_3$ | 0.9205 |
| $\pi_1 - \pi_2\pi_4$ | 0.8304 |
| $\pi_1 - \pi_3\pi_4$ | 0.8798 |
| $\pi_1 - \pi_2\pi_3^{-1}$ | 0.9495 |
| $\pi_1 - \pi_2^{-1}\pi_3$ | 0.8556 |
| $\pi_1 - \pi_2\pi_4^{-1}$ | 0.8304 |
| $\pi_1 - \pi_2^{-1}\pi_4$ | 0.8096 |
| $\pi_1 - \pi_3\pi_4^{-1}$ | 0.8798 |
| $\pi_1 - \pi_3^{-1}\pi_4$ | 0.9532 |
| $\pi_1 - \pi_2\pi_3\pi_4$ | 0.9065 |
| $\pi_1 - \pi_2\pi_3\pi_4^{-1}$ | 0.9065 |
| $\pi_1 - \pi_2\pi_3^{-1}\pi_4$ | 0.9624 |
| $\pi_1 - \pi_2^{-1}\pi_3\pi_4$ | 0.8556 |
| $\pi_1 - \pi_2\pi_3^{-1}\pi_4^{-1}$ | 0.9499 |
| $\pi_1 - \pi_2^{-1}\pi_3^{-1}\pi_4$ | 0.9590 |
| $\pi_1 - \pi_2^{-1}\pi_3\pi_4^{-1}$ | 0.8556 |

### 2.4. Validation: Numerical and Field Experiments

The target dispersion coefficients, channel geometry, and hydrodynamic data required for the proposed equation were obtained from the numerical validation data sets. For the same purpose five different pool-riffle macrostructures were identified within the Bellavista river, located in the city of Tomé, Biobío Region, Chile (Figure 4) and tracer experiments were presented from each sequence. Each study site consisted of an upstream riffle, jet inflow, pool, tailout, and downstream riffle, all of which presented straight channel alignment. The pool-riffle macrostructures studied here were characteristic of mixed sand-gravel streams.

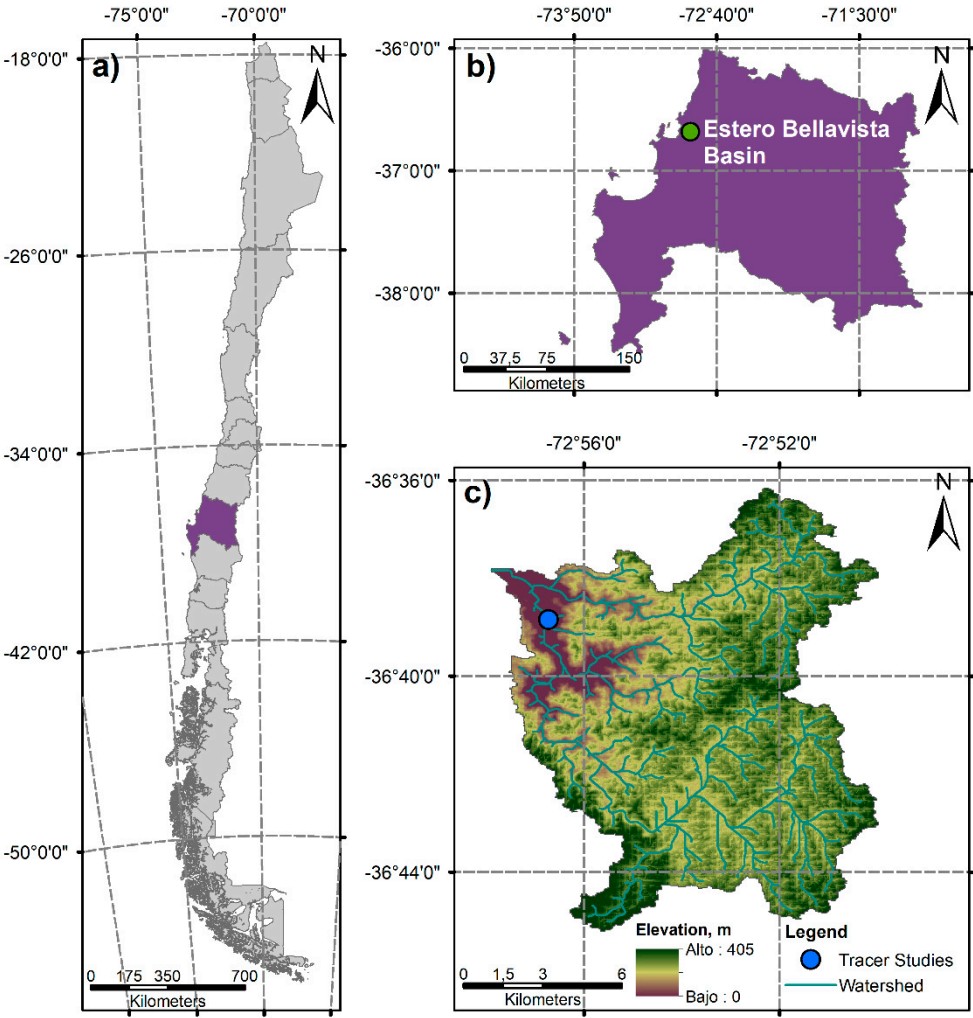

**Figure 4.** (**a**) Location of the Biobío region in Chile; (**b**) the Bellavista watershed location within the Biobío region; and (**c**) the tracer study location within the Bellavista watershed.

Field tracer experiments used a saline (NaCl) solution as a tracer through pool-riffle macrostructures, a method which was previously used to estimate discharge [60], assess hydraulic dead zones [61], and estimate travel times [62]. The experiments performed in the Bellavista river included an instantaneous injection across the entire river width, at the upstream riffle (i.e., to ensure instantaneous mixing conditions). Two sampling points were located upstream and downstream of the deepest point in the pool, following the central thalweg line. Table 4 shows the geometric and flow characteristics for each study reach. Couplets of tracer concentration-time curves were sampled every 30 s and the longitudinal dispersion coefficient was estimated using the TM and MM methods.

**Table 4.** Geometric and flow characteristics for each study reach.

| Variable | Bathymetry Scenario | | | | |
|---|---|---|---|---|---|
| | Bellavista 1 | Bellavista 2 | Bellavista 3 | Bellavista 4 | Bellavista 5 |
| $h_{pt}$ (m) | 0.48 | 1.17 | 0.39 | 0.81 | 0.57 |
| $h_{rt}$ (m) | 0.14 | 0.09 | 0.14 | 0.07 | 0.05 |
| $\Delta_z$ (m) | 0.21 | 0.96 | 0.12 | 0.57 | 0.39 |
| $L_{PR}$ (m) | 15.81 | 11.30 | 14.97 | 8.10 | 11.82 |
| $B_r$ (m) | 8.61 | 6.46 | 16.40 | 5.20 | 12.00 |
| $B_p$ (m) | 7.25 | 5.80 | 15.10 | 4.50 | 8.22 |
| $u^*\left(\frac{m}{s}\right)$ | 0.16 | 0.26 | 0.15 | 0.30 | 0.21 |
| $U\left(\frac{m}{s}\right)$ | 0.38 | 0.51 | 0.42 | 0.26 | 0.30 |
| $S_w$ (−) | 0.008 | 0.010 | 0.008 | 0.020 | 0.010 |
| $S_B$ (−) | 0.086 | 0.058 | 0.086 | 0.086 | 0.319 |

Results from the numerical validation set and field experiments were compared with the longitudinal dispersion coefficients calculated using the proposed equation. Additionally, results were evaluated against the empirical formulas described in Table 1.

## 3. Results

The simulated concentration time curves for a constituent passing through the pool-riffle macro-structure in scenario 1 are shown in Figure 5. There was one curve per observation point illustrating the gradual decrease in peak concentration along the pool-riffle sequence. Integrating the concentration over time showed that the mass of each curve maintained a constant value of $15\,\mathrm{g\,s\,m^{-3}}$, fulfilling conservation of mass within the model domain. Additionally, the shape of the curves indicated that the constituent was normally distributed, owing to its uniform distribution at the upstream boundary, with increasing variance, as it progressed downstream.

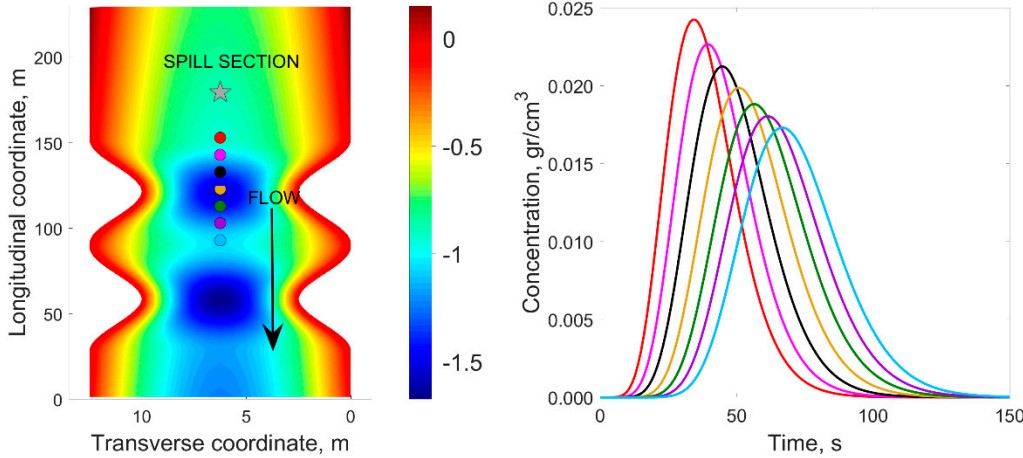

**Figure 5.** (**a**) Plan view of the sampling points in the synthetic bathymetry domain for the time concentration curves. Pont spacing was 10 m; and (**b**) time-concentration curves for bathymetric setting with $\Delta_z = 0.494$.

Longitudinal dispersion coefficients were calculated from the simulated concentration time curves for each one of the considered synthetic bathymetry scenarios, using the TM and MM methods. Both TM and MM methods estimated values of the dispersion coefficient of the same order of magnitude (Figure 6), which were reflective of small streams and canals. It was also clear that the behavior of the curves was similar in shape, reaching a minimum around a residual pool depth of 0.4 m. The dispersion

coefficient in both methods showed an inverse relation to the hydraulic radius through the residual pool depth.

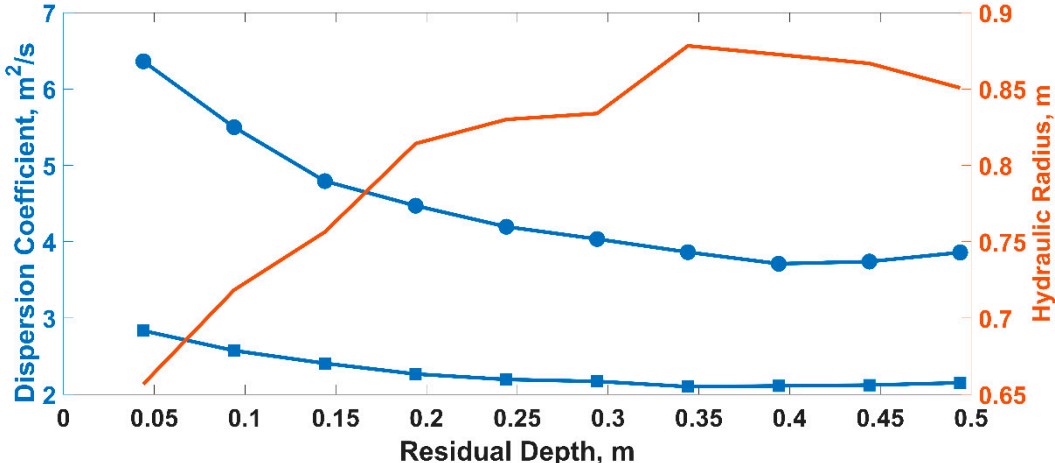

**Figure 6.** Comparison of the method of Thomann and Mueller [54] and the method of the moments [55] for estimation of the longitudinal dispersion coefficient for the bathymetries described in Table 2. Variation of the dispersion coefficient (blue) and the hydraulic radius in the pool (orange) with respect to the residual pool depth for (**a**) method of Thomann and Mueller (box); and (**b**) method of the moments (circles).

To explore the relation between turbulent fluid motion through complex bathymetry and its impact on the dispersion coefficient, Figure 7 shows the simulated range of variation in turbulent kinetic energy (TKE) for each one of the studied pool-riffle macrostructures. The magnitude of the TKE variation presented a minimum range between the residual pool depths of 0.14 and 0.20 m, with increasing variance in both directions. As the residual pool depth increased, an increased variance in TKE occurred, owing to the sudden fluid acceleration/deceleration, as the flow entered and exited the pool. The median values of TKE showed a decreasing trend with increasing residual pool depth, with a minimum value around 0.4 m. It could be noted that the trend in median TKE values presented the same tendency as the dispersion coefficient, reinforcing the direct relation between the dispersion coefficient and fluid turbulence. These findings support the appropriateness of the three-dimensional numerical modeling results.

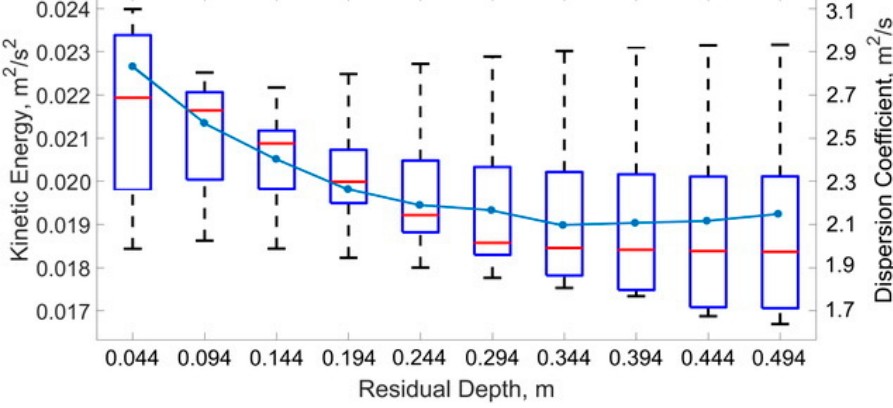

**Figure 7.** Variation of turbulent kinetic energy for each scenario (expressed as the residual pool depth).

Table 5 shows the 16 input sets used in the sensitivity analysis and applied to the 3D transport model, for Scenario 1. Results indicated that smaller initial dispersion values resulted in a greater difference between the initial and estimated dispersion coefficient. This difference decreased as initial

coefficients increased, suggesting that the estimated dispersion coefficients were more sensitive to smaller initial values. Results were within an order of magnitude, which was an improvement over the predictions from existing empirical relations, when considering the bed macrostructures. Additionally, the null dispersion scenario resulted in a positive estimated longitudinal dispersion. This implied the presence of other dispersive processes that related to an unknown combination of the complex flow field within a pool-riffle macrostructure and numerical diffusion. Collectively, these processes resulted in dispersion on the order of 0.44 m$^2$/s, which was expected to be present throughout all simulations.

**Table 5.** 3D model initial and estimated dispersion coefficients based on the method of moments in m$^2$/s.

| Initial Dispersion Coefficient | Estimated Dispersion Coefficient (Method of Moments) | Difference |
|:---:|:---:|:---:|
| 0.0 | 0.437 | 0.437 |
| 0.5 | 1.707 | 1.207 |
| 0.6 | 1.789 | 1.189 |
| 0.7 | 1.872 | 1.172 |
| 0.8 | 1.963 | 1.163 |
| 0.9 | 2.055 | 1.155 |
| 1.0 | 2.150 | 1.150 |
| 1.1 | 2.229 | 1.129 |
| 1.2 | 2.307 | 1.107 |
| 1.3 | 2.385 | 1.085 |
| 1.4 | 2.463 | 1.063 |
| 1.5 | 2.541 | 1.041 |
| 1.6 | 2.619 | 1.019 |
| 1.7 | 2.697 | 0.997 |
| 1.8 | 2.776 | 0.976 |
| 1.9 | 2.854 | 0.954 |
| 2.0 | 2.932 | 0.932 |

Results from the dimensional analysis indicated that $\pi_1$ as function of $\pi_2\pi_3^{-1}\pi_4$ presented the highest correlation value ($R^2 = 0.9624$) and was, therefore, proposed as the predictive equation. The proposed equation for the longitudinal dispersion coefficient through pool-riffle macrostructures is reported in Equation (8).

$$D_x = \left(0.4876\frac{U}{u_*}\frac{S_B}{S_w} + 8.3683\right)H\,u_* \tag{8}$$

Mathematically, Equation (8) was only valid when $S_w \geq 0$; however, $S_w \leq 0$ was physically quite unlikely since pool-riffle structures would have a non-zero water surface slope from the upstream riffle crest to the downstream pool. Additionally, pool-riffle sequences only form in relatively steep fluvial environments with $0.002 < Sw < 0.04$, where Equation (8) is applicable [63,64].

Equation (8) includes reach-averaged variables similar to those proposed by earlier empirical equations in the literature, but, directly considers the vertical and horizontal expansion–contraction geometry of pool-riffle macrostructures through the terms $S_B$ and $S_W$. For example, in a case where there is no horizontal expansion, $S_B$ is zero (i.e., there is no pool-riffle structure), Equation (8) reverts to the simplified version of the equation proposed by Elder [28] for prismatic channels. It is interesting to note that subsequent research found Elder's equation (Table 1) to underestimate the value of the longitudinal dispersion coefficient [26,32,34,42]. When the proposed Equation (8) was simplified to the form of Elder's equation ($S_B = 0$), the coefficient within the parentheses was less subject to underestimation as it predicted a 41% greater longitudinal dispersion coefficient.

The validation of Equation (8) was carried out using the remaining three numerical validation scenarios and the five field tracer experiments. Comparison between the predicted longitudinal dispersion coefficients and those in the validation data is shown in Figure 8 The coefficient of determination for Equation (8) is $R^2 = 0.64$, providing an acceptable level of overall accuracy.

This implied that by including pool-riffle expansion ratios in addition to the reach average values, 64% of the variability was explained by Equation (8). The resulting equation was an improvement from previous work that did not include bed complexity, which reported $R^2$ values of 0.55, 0.25, 0.5, and 0.55 [14,26,29,31]. In addition, 75% of the data were within the acceptable range defined by Seo and Cheong [26], which was higher than the 34%, 47%, and 31% obtained by Antonopoulos et al. [65].

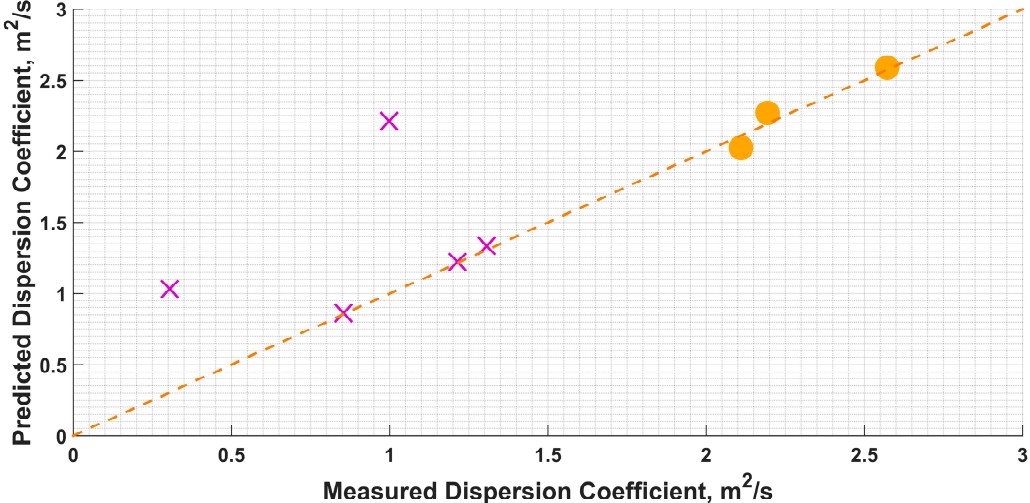

**Figure 8.** Comparison of values predicted by Equation (8) and those in the validation set. The orange points represent the field experiments and the purple x represent the numerical values.

The two points located farthest from the predictive line were the two most extreme ends of the morphological spectrum examined in the field tracer tests. These points represented field experiments 2 and 3, with the largest and smallest relative pool depths, in addition to the lowest ratio of $B_r/B_p$ from the field data. Therefore, these were the least well-developed pool-riffle macro-structures that were evaluated in the field sites. From the available data it is uncertain why this occurred, but both of these points fell within a predictive factor of 2. This indicated that further data near the extremes of the analyzed geometries needed to be generated to refine the coefficients in the proposed Equation (8).

## 4. Discussion

The predicted dispersion coefficients in this study ranged between 0.86 and 2.59 m$^2$/s (Table 6), which were substantially less than those reported in natural rivers [31,35]. This was because the processes that were being captured in the 3D numerical model corresponded to a single bed macro-structure that did not include components like sinuosity, streambank dead zones, and vegetation that were usually lumped together in reach-averaged equations. Hence, these results represented the physically-based processes through an isolated pool-riffle sequence. This range of values indicated a minimum level of influence imposed by pool-riffle macroforms for longitudinal mixing processes. When examined against data from real rivers [26,33] that are known to have pool riffle structures, the influence of bed macrostructures calculated herein could constitute up to 100% of the dispersion values seen in small rivers and less than 2% in larger streams.

Specifically, it could be seen that the bed macro-structure imposed changes to the flow field as it progressed through the pool-riffle feature, with resultant impacts on the dispersion coefficient. Recent work suggests that the longitudinal dispersion coefficients should be directly related to the geometric properties of the river, specifically a direct relation to the hydraulic radius [35]. However, this study identified an inverse relationship (Figure 6), suggesting that the dispersion coefficient was minimal when the hydraulic radius was maximal. The numerical model results illustrated that this occurred because as the flow depth and hydraulic radius increased through the deepest part of the pool, there was an increase in flow divergence, decrease in velocity, and overall reduction in advective

transport. The reduced advection was directly related to the dispersion coefficient and resulted in a decreased value. Additionally, the bed macroform affected the turbulent fluid exchange, owing to divergence through the pool and convergence over the riffle, which has been found to be related to the formation of unstable turbulent structures [66]. As dispersion corresponds to the scattering of particles due to turbulent fluid movement, it was suggested that the variation of the dispersion coefficient must be linked to turbulence [29]. Results showed that as flow divergence occurred in the pool and velocity decreased, the distribution of TKE responded with a proportional decrease (Figure 7). This reduction in TKE and fluid exchange effectively reduced the dispersion coefficient supporting previous research in the literature [6,54,56–58,66].

**Table 6.** Longitudinal dispersion coefficients. $D_m$ are the coefficients from numerical modeling and field experiments, $D_p$ are the coefficients predicted by Equation (8).

| Bathymetry Scenarios | $D_m$, m$^2$/s | $D_p$, m$^2$/s |
|---|---|---|
| 3 | 2.110 | 2.025 |
| 6 | 2.194 | 2.270 |
| 9 | 2.572 | 2.589 |
| Bellavista 1 (Tracer experiments) | 1.214 | 1.222 |
| Bellavista 2 (Tracer experiments) | 0.305 | 1.032 |
| Bellavista 3 (Tracer experiments) | 0.998 | 2.212 |
| Bellavista 4 (Tracer experiments) | 0.854 | 0.862 |
| Bellavista 5 (Tracer experiments) | 1.306 | 1.335 |

Equation (8) attempts to characterize the response of the dispersion coefficient to the complex flow field through bed macro-structures seen in real rivers. For context, the equations presented in Table 1 were applied to the numerical results and evident differences were apparent in the predicted dispersion coefficients (Figure 9). The box plots showed the wide range of values in the predicted dispersion coefficients spanning two orders of magnitude, when applied to pool-riffle macrostructures. Theoretically, these variations could be attributed to their variable formulations (e.g., non-dimensional numbers like Fr, B/h, U/U*, etc.) and the morphology of the streams (or flumes) that they were developed for. For example, Zeng and Huai [34] achieved good results for trapezoidal channels, yet the same formulation underestimated the dispersion coefficient values for rectangular channels. Such differences illustrated that channel morphology is a principle component of the variation that we see in the theoretical formulations and should be included more directly, in the estimates of the dispersion coefficient.

The proposed methodology to develop process-based mixing equations based on virtual morphological features provide an exciting new avenue of research. This methodology allowed for the simulation of specific morphological features and characterization of their individual impact on the hydrodynamic mixing processes. Future work using simulated river bathymetry and numerical modeling can identify the physically-based influence of other morphological structures like bars, width expansion/contraction, step-pools, sinuosity, etc., on river mixing. Ultimately, these individual dispersion relations would allow various combinations of features to be combined in reach-averaged values, via a partitioning approach seen in other branches in hydraulics, such as roughness and shear stress [67–70]. Current work is exploring the combinations of geomorphic features to guide restoration activities.

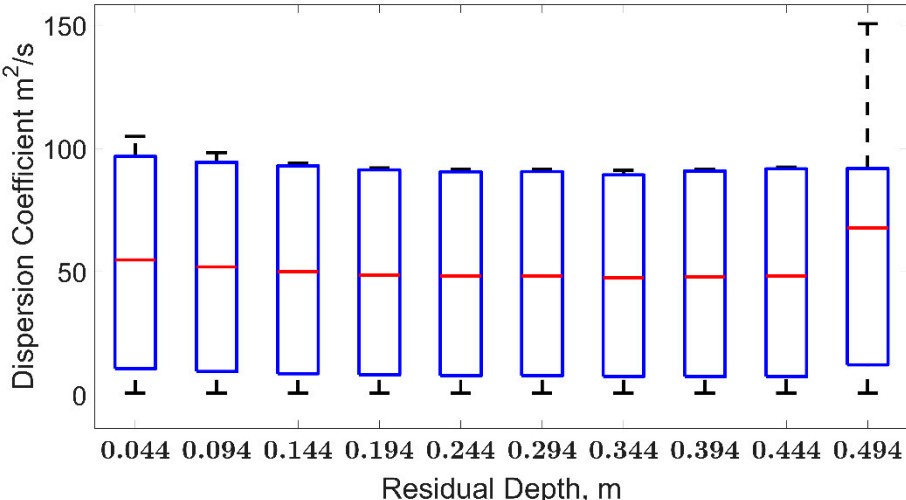

**Figure 9.** Variation of dispersion coefficient values calculated in the numerical pool-riffle sequence through the empirical equations of Table 1, for each scenario.

## 5. Conclusions

Reliable estimation of mixing in fluvial systems is critical for evaluating constituent and pollutant transport within rivers. Different equations applied to the same hydraulic conditions through a pool-riffle macroform can result in dispersions coefficients ranging in two orders of magnitude. The incorporation of local morphological variables and dimensional analysis for predicting longitudinal dispersion coefficients, like that proposed in this study, can reduce the variability in predicted results to within a factor of two. Therefore, an improved equation for calculating the longitudinal dispersions coefficient that accounts for flow expansion and the contraction seen in pool-riffle macrostructures is presented in Equation (8). This equation is proposed for application in one-dimensional water quality models for reaches of rivers with pool-riffle macrostructures. However, the use of the equation should be evaluated more extensively, in order to understand the valid ranges for its use.

More importantly, this study showed that employing a 3D hydrodynamic and transport model provided a process-based understanding of hydrological mixing through a single morphological macroform. In combination with synthetic bathymetry, these models would allow for direct evaluation of other channel features like alternating bars, varying sinuosity, and streambank vegetation and their respective significance in the mixing and transport phenomena. Such an approach could be further applied to engineering structures such as agricultural diversions, bank protection, and embankments, to understand the impact of river infrastructure. This approach is particularly powerful when combined with dimensional analysis for the identification of more robust process-based results.

**Author Contributions:** Conceptualization, P.F.-A., D.C., H.A. and A.T.; Methodology, P.F.-A., D.C. and H.A.; Software, P.F.-A.; Formal Analysis, P.F.-A.; investigation, P.F.-A. and D.C.; Data Curation, P.F.-A.; Writing—original draft preparation, P.F.-A. and D.C.; Writing—review and editing, D.C., H.A. and A.T.; Visualization, P.F.-A.; supervision, D.C. All authors have read and agreed to the published version of the manuscript.

**Funding:** This research received no external funding.

**Acknowledgments:** The authors wish to acknowledge two research grants that together financed this research, project MECESUP USC1795 from the Ministerio de Educación de Chile and project DINREG 07/2018 from the Dirección de Investigación de la Universidad Católica de la Santísima Concepción, Chile.

**Conflicts of Interest:** The authors declare no conflict of interest.

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
