# Peer review of "The Influence of Pool-Riffle Morphological Features on River Mixing"

_water, doi:10.3390/w12041145_

Round 1

Reviewer 1 Report

This study addresses the interesting topic of calculating a predictive longitudinal mixing equation to model the influence of pool-riffle sequences. The authors used DELFT3D with synthetic bathymetry to conduct numerical experiments, and then validated the predictive equation they proposed. Overall, the authors did a good job and the results are clearly presented. Though I'm not very familiar with pool-riffle features, I'm specialized in 3-D hydrodynamic modeling. I have some comments on the numerical modeling part, and I hope authors would take some time to address them:   1. Can you explain the use of a uniform Manning number (0.025) over this entire modeling domain? 2. Since you are using an unstructured grid setting, the horizontal and vertical spatial resolutions play important roles in your model results. But I didn't see much about how you chose the specific resolutions and how these choices could affect your results. 3. Figure 5: Can you make squares a bit larger so it can tell more clearly from the circles? 4. Line 300: Which metrics did you use to reach the conclusion ("accuracy is high")? Did you calculate R^2 or any correlation coefficient?

Reviewer 2 Report

Reviewed paper deals with the investigation of the Influence of Pool-Riffle Morphological Features on River Mixing. The subject of the paper well fits the scope of Water. The paper contains new interesting results and could be considered for publication in Water after revision taking into account the following comments:

  1. In the reviewed paper, a very important and at the same time very delicate issue of parameterization of the longitudinal dispersion coefficient in channel flows is considered. Longitudinal dispersion in channel flows is of great practical as well as significant theoretical interest. The main practical interest is the assessment of the linear dimensions of the pollution zones during the volley of pollutants into the watercourse . However, the assessment of the effective value of the dispersion coefficient is much more complicated, since its value in the transverse and even more significant in the longitudinal direction is determined by the structure of the turbulent flow and Taylor diffusion (A.S. Monin, A.M. Yaglom. Statistical Fluid Mechanics, Volume I: Mechanics of Turbulence (Dover Books on Physics) related to the non-uniformity of the velocity field under consideration over the flow cross section:

        Dxx ~ (1/H) 0H dz/Kzz(z) lH((V(l)-Vср)dl)2

        Moreover, the value of this coefficient can significantly exceed the                value of the coefficient of turbulent diffusion. Strictly speaking, this              expression can be applied only in the case of uniform turbulence, for a          flow with a constant average velocity. At the same time, natural                  watercourses, as a rule, are characterized by rather complicated                  irregular morphometry. In this connection, the question arises                      whether the process of the longitudinal transport of a pollutant can be          described in the framework of the usual Fick model of diffusion, and            whether a transition to generalized diffusion models is required.                  Unfortunately, the paper does not provide a theoretical analysis of the          dispersion in the flow with a non-uniform distribution of both                        transverse and longitudinal velocity fields, which are determined by              the inhomogeneity of the flow morphometry.

  1. In addition to those indicated in the reviewed article, a large number of empirical relationships are known for estimating the coefficient of longitudinal dispersion. Their characteristic feature is a very wide scatter at the same values of the governing parameters. This introduces the question of the correctness of applying Fick diffusion models to watercourses with complex morphometry. Therefore, it would be useful if the authors of the reviewed paper analyzed the factors which cause such a wide spread in the estimates of the coefficients of longitudinal dispersion for various empirical models.
  2. When solving the problem of taking into account the influence of heterogeneity of river flow morphometry on the longitudinal dispersion coefficient, the ratio of the characteristic lengths of the computational domain Lp and the size of the inhomogeneity LM, is of fundamental importance. At LM/Lp << 1, diffuse models can probably will be correct, while at LM/Lp ~ 1 they can hardly be correct.
  3. Fick model assumes and determines the “normality” of the distribution of the pollutant cloud in the flow, which is well justified for a uniform flow. In a flow with complex morphometry, the flow is not steady-state; accordingly, the “normality” of the distribution of the pollutant cloud will be violated. In this case, using the Fick diffusion model for description seems incorrect.
  4. It should be taken into account that with complex morphometry of the watercourse, the inertial terms become significant and the flow hydrodynamics, including the longitudinal transport of pollutants, become significantly complicated.
  5. The proposed relation for estimating the longitudinal dispersion based on the morphometric characteristics of the channel (relation (8)) gives known formula Dx ~ 8.3683 H V* in the case when SВ≡0. What about the case Sw≡0 ?
  6. Taking into account the high error in determining the main initial parameters of the transport model, it is hardly justified to set the values of the coefficients of longitudinal dispersion, indicating the values of the calculated coefficients with five significant digits after the decimal point.

Round 2

Reviewer 2 Report

The authors answer my comments in proper way. I recommend this paper for publication.